# Collaboration through Integrated BIM and GIS for the Design Process in Rail Projects: Formalising the Requirements

**Sahar Kurwi, Peter Demian \*** , **Karen B. Blay and Tarek M. Hassan**

School of Architecture, Building and Civil Engineering, Loughborough University, Loughborough LE11 3TU, UK; S.Kurwi@lboro.ac.uk (S.K.); K.B.Blay@lboro.ac.uk (K.B.B.); T.Hassan@lboro.ac.uk (T.M.H.)
\* Correspondence: P.Demian@lboro.ac.uk

**Abstract:** Many of the obstacles to effective delivery of rail projects (in terms of cost, time and quality) can be traced back to poor collaboration across complex design teams and supply chains. As in any infrastructure delivery process, it is important to make decisions collaboratively at an early design stage. Advanced systems such as Building Information Modelling (BIM) and Geographic Information Systems (GIS) can facilitate collaboration during the decision-making process and boost work efficiencies. Such potential benefits are not realised because the roles of BIM and GIS in facilitating collaboration are not clearly understood or articulated. This paper aims to identify and articulate collaboration requirements during the design stage of rail projects. To achieve this, a mixed-method approach was employed to examine the issues that hinder collaboration in rail projects. An online questionnaire was designed to assess the state-of-art in BIM and GIS, followed by fifteen follow-up face to face interviews with experts to identify collaboration issues and suggestions to overcome them. The research identified the main challenges to effective collaboration and provided suggestions to overcome them. The main challenges were managing information and a reluctance to use new collaboration technologies. The main solution which emerged from the data was to develop an original Collaborative Plan of Work (CPW). The developed CPW is tailored to rail projects and has been formulated by combining the RIBA (Royal Institute of British Architects) Plan of Work and the GRIP Stages (Governance for Railway Investment Projects). This comprehensive plan of work, which is uniquely collaboration-focused, is significant because it can be further developed to formulate a precise process model for collaboration during the design process of rail projects. Such a process can (for example) be configured into the workflow prescribed by a Common Data Environment.

**Keywords:** rail projects; collaboration; GIS; BIM; RIBA Plan of Work; GRIP stages





## 1. Introduction

In recent years, the importance of collaboration in the construction sector has driven innovations in its Information and Communication Technologies. Collaboration across parties and processes facilitated by digital technologies is often identified as a key necessity to deliver successful rail projects. For example, a Network Rail [1] report forecasted that the adoption of the collaborative working practices facilitated by digital technologies could deliver savings of 30%. Collaboration is crucial for any project to achieve the project objectives in terms of cost, time and quality, but is particularly important in rail projects due to their added complexity and scale.

The complexity of rail projects is indeed noteworthy. Like many linear infrastructure projects, they can span hundreds of kilometres, covering diverse geographies and environments. Rail projects carry the added complexity of extensive line-side furniture. This complexity is reflected in the multi-layered information to be exchanged during these projects and the numerous stakeholder groups needing to communicate [2].

Collaboration obstacles often ultimately amount to issues of information management: providing the right information to the right person at the right time to accomplish tasks.

This sentiment repeatedly occurred in discussions with practitioners throughout this research and is also frequently quoted by researchers. One study [3] cites it for making human-centred computer systems more context-aware. Another study [4] invokes this notion as a driver for exploiting location data on Twitter, as a social medium, to direct health information. In the building design domain, Zanni et al. [5] refer to this sentiment for integrating sustainability into the building design process in a more coordinated way. To tackle such information management challenges, construction professionals would need a clear model of the collaboration process, showing key information exchanges and decision points. This paper attempts to fill this gap by developing a Collaborative Plan of Work (CPW) by combining the RIBA Plan of Work and GRIP Stages, to serve as a comprehensive guide to collaboration in the design stage of rail projects. This CPW focuses on the collaboration process and the underlying information management. Based on previous research [6], BIM and GIS are positioned as the pertinent information technologies needed to support collaboration in rail design as part of the CPW.

Such emphasis on collaboration and information management is lacking in existing process models. The RIBA Plan of Work 2013 was infused with added flexibility and customisability to bring integration to the project team [7]. It was presented as a toolkit rather than a rigidly prescriptive process. Poor coordination and design team fragmentation were addressed merely by suggesting the use of emerging technologies such as BIM and GIS. Nonetheless, within the process set out in the RIBA Plan of Work, precise details of how to collaborate and how to use collaboration technologies (such as BIM and GIS) are absent ([8], p. 54). The RIBA Plan of Work 2020 addresses sustainability issues more directly, with the drive towards net-zero projects. However, the "Information Exchanges" taskbar is not significantly developed leading up to the 2020 version. There remains a need for more explicit guidance on the collaborative design process, the underlying information management and the use of technologies such as BIM and GIS to facilitate this collaboration.

BIM and GIS are the two technologies chosen in this research to underpin a collaborative environment. The starting point is BIM. "Fully collaborative 3D BIM" (what used to be referred to as Level 2 BIM maturity in the perhaps now superseded Bew-Richards BIM maturity model, [8]) has been mandatory for UK public sector projects since 2016. To realise the full potential of BIM, it is essential to integrate it with GIS as a complementary technology. BIM's strengths are focused on modelling indoor space scales, whereas (pertinent for rail projects) GIS focuses on outdoor space scales [9]. Wang et al. [10] refer to these as micro-level and macro-level representations, respectively. BIM lacks GIS' ability to analyse spatial data [11]. Given their complementary functions, researchers [12,13] have reported the synergy which can be realised from exploiting their respective strengths and combining them in an integrated way.

Notwithstanding the discussion above of the RIBA Plan of Work, much BIM research focuses on technology aspects, at the expense of people and process aspects [14]. Elsewhere, it has been reported that for successful implementation of collaboration systems, people and process issues are much more important than technology aspects [15]. There is a lack of coordination among people, tools, deliverables, and information requirements [16]. The BIM process requires new processes and communication channels [17] for providing the accurate information needed throughout the lifecycle of the constructed facility. These processes and communication channels need to support the unstructured "messy talk" which often arises due to the complexity of collaboration on design and construction projects [18]. Therefore, this paper aims to identify and articulate the requirements for effective collaboration among project participants in rail projects.

Through a questionnaire survey and follow-up in-depth interviews, *collaboration* requirements in the rail project design process are distilled into a plan of work for collaborative design. First, a literature review of research on collaboration and its requirements is presented. Secondly, the research methods used are described, then the findings are presented and discussed. Finally, conclusions are drawn, and future work recommended.

## 2. Literature Review: Collaboration Requirements and the Potential of BIM and GIS

The general business benefits of collaboration, particularly that mediated by software systems, has long been reported by scholars [19]. Researchers [20] from their state-of-the-art review, identified the constituent activities that are included in the term "collaboration": sharing information, coordinating tasks and conflict resolution. Specifically, in the construction sector however, tools and systems for collaboration have been driven by "what to implement" with little focus on the "how". As such, several technologies such as Common Data Environments, videoconferencing and electronic whiteboards have been developed, but their success rate has been minimal [21]. Case studies of IT adoption in projects reveal that, for effective collaboration, the focus should be on "how to implement" rather than "what to implement" [21]. Therefore, there is a need to identify the requirements for effective collaboration and to articulate precise steps of how to collaborate.

Collaborative work is a core theme of the UK Government strategy. By mandating BIM as an enabler of more efficient, collaborative working, collaboration was positioned to deliver significant savings in the design and procurement stages of public sector projects [22]. Aligned with UK Government policy, professional bodies in the UK have also singled out collaboration as a priority. For example, the Institution of Civil Engineers' industry-led Project 13 framework includes an "organisation" pillar and identifies collaboration across supply chains as one of the requirements for delivering and managing high-performing infrastructure [23].

Collaboration can address common problems such as clash detection [24,25], rework, and better decision making [26]. Collaboration has been found to facilitate the sharing of knowledge and skills, particularly in multidisciplinary design [27], to help maintain relationships [28] and to reduce risks [29].

Research focusing on collaborating during the design phase of construction projects [30] has identified "super soft" critical success factors such as passion and enthusiasm in individuals and shared values amongst design team members. Similarly, researchers [31] studying design collaboration in university student teams rather than practitioners, observed that collaboration technologies appeared to moderate effects of ownership and competition in individuals. Such factors are implicitly precursors to design collaboration and have important implications for enabling collaboration, but the intricacies of the collaborative design process remain unexplored, particularly for rail projects. The process of collaboration is aligned to wider processes of design and construction, which have been extensively researched in the context of BIM. In his BIM framework, Succar [32] includes a "process field", as an important area of research for delivering constructed facilities using a more effective, BIM-based process. Collaboration is an important component of this process. It is noteworthy that BIM for infrastructure lags behind BIM for buildings. One study [33] presents I-BIM, an information management systems specifically for linear infrastructure, particularly rail. Although they highlight the challenge of parametric modelling linear objects, they articulate the value of placing BIM data in a geographic context.

Supplementing BIM with the complementary functionality of GIS is a promising basis for supporting collaboration in the design phase of rail projects. Wang et al. [11] report the results of a bibliographic analysis of research into BIM-GIS integration specifically for sustainability. They highlight the challenge of data integration between the two platforms and explore different levels of "leadership" between BIM and GIS. Zhu at al. [34] focus on data integration. They identify IFC and CityGML as the dominant formats for BIM and GIS respectively. They note that there is no single ontology comprehensively encompassing both platforms, but that data exchange is possible on a project-by-project basis. Fosu et al. [35] similarly review literature on BIM-GIS integration for asset management and conclude that such integration had not quite made the transition from academic research to industrial reality. Abd et al. [36] depend on commercial platforms to integrate BIM and GIS. They describe a process of using Autodesk InfraWorks to exchange information between a BIM in Revit and a GIS model in ArcGIS. Ma and Ren [37] review 42 publications covering BIM and GIS integration in the context of smart cities; they report that the manual

export/import of data from one platform to the other seems to be the main mechanism for BIM-GIS integration, and observe more frequent application to buildings rather than infrastructure. Shr and Liu [38] apply BIM-GIS integration to the repair of the railway infrastructure and use a standard database system as the bridge between the two platforms. Amirebrahimi et al. [10] apply BIM-GIS integration to flood damage assessment, noting the complementary need of data from both platforms for that application. From their literature review, Liu et al. [39] adopt Amirebrahimi et al.'s [10] categorisation of efforts to integrate BIM and GIS into data-, process- and application-level integration, and go on to review the application of this integration to tasks peripheral to collaboration, such as cadastre and safety. Irizarry et al. [40] present a protype system which integrates BIM and GIS for supply chain management, visualising the flow of materials and resources linked to a building model on GIS-based maps. Vacca and Quaquero [41] present a workflow for integrating GIS and BIM data via respective information standards for each platform. The application of the workflow to two case studies demonstrates the feasibility of information exchange between BIM and GIS. In recent UK industry practice, BIM and GIS were both applied to the major rail project, Crossrail [42]. Crossrail used a Common Data Environment called Enterprise Bridge which allowed all project data to be shared in a single central location. The project's BIM protocol did specify the use of GIS as a supplement to BIM, notably to provide context to the various models.

The overarching theme emerging from the literature review is that the biggest challenge to enabling effective collaboration is the lack of coordination among people, tools, deliverables, and information requirements [16,43]. Integrating BIM with GIS can provide a complete toolset to support collaboration between participants throughout the lifecycle of a rail project. Furthermore, a clear plan of work for the railway which sets out information exchanges and roles in the design process would facilitate this collaboration and inform the integration of BIM and GIS as a holistic software platform to enable it.

## 3. Research Method

A mixed-methods approach was adopted to examine collaboration issues and identify collaboration requirements. Data collection began with an online questionnaire survey when more than 500 survey invitations were sent by email or social media to construction professionals with expertise in rail/BIM/GIS. The purpose of the questionnaire was to establish the current status of BIM and GIS in railway design, focusing on their use as collaboration enablers during railway design. A total of 114 responses were received from the 500 invitations, giving a 23% response rate. The questionnaire was followed up by 15 in-depth interviews to investigate further the issues that were highlighted by the questionnaire respondents and solicit potential suggestions to overcome these issues.

The questionnaire instrument followed the funnel approach [44] in which the questionnaire starts with very broad questions and narrows down the scope of the questions reaching the end with a very specific focus. The interview instrument consisted of three sections. The first section further explored collaboration issues in railway design emerging from the questionnaire data; the second section solicited the interviewee's views on the potential of BIM and GIS to address these collaboration issues and collected other suggestions for overcoming them; the third section questioned the interviewee about their background and employer.

A non-probabilistic, purposive sampling approach was adopted to send questionnaire invitations based on the experiences of participants. This constitutes Expert Sampling [45], whereby a sample of persons with known or demonstrable experience and expertise in the area is selected. The questionnaire participants were chosen based on their backgrounds and specialisations in different companies.

The sampling for the interviews followed the same approach as for the questionnaire. The selection of interview participants was based on three factors: the respondent's willingness to be involved in an interview; their experience in BIM, GIS and rail and their use of BIM/GIS for collaboration. Fifteen out of fifty people accepted the invitation to

be interviewed (a response rate of 30%). The length of each interview was in the range 1–2 hours. Interviews were audio-recorded and transcribed.

Assuring validity and reliability of instruments is a challenge in collecting and analysing survey data (whether questionnaire or interview surveys). The communication medium used can have a significant effect. For example, de Vaus [46] argues that face to face data collection reduces the bias in responses. This research used self-administered online questionnaires. Interviews were conducted face to face where possible (three of 15), with the remainder conducted via videoconferencing tools. Data collection instruments were developed with particular attention to validity. Internal validity is the degree to which a measure accurately represents an observed variable. Ensuring internal validity starts with an understanding of what is to be measured and then making the measurement as correct as possible. Reliability is the extent to which a variable or a set of variables is consistent in what is intended to be measured, so that multiple repeated measurements give the same results. One indicator of reliability is the consistency of measurements taken from the same individual at two points in time; pilot rounds of questionnaire data collection (not used in the main analysis) provided some reassurance of the reliability of the questionnaire instrument.

## 4. Data analysis and Discussion

### 4.1. Questionnaire Data Analysis

SPSS was used to apply descriptive statistics and hypothesis testing to the questionnaire data. Most questions used the Likert scale, acknowledging the arguments for and against Likert scales and how to analyse Likert data [47]. Applying parametric statistics methods to Likert data has been both criticised and favoured by researchers [48]. For the questionnaire used here, the 5-point scale was adopted; studies [49] have shown that the differences in data characteristics are not significant between 5-, 7- and 10-point Likert scales. This research used a 5-point scale, where responses were scored from -2 ("Strongly Disagree" or "Not Important") to +2 ("Strongly Agree" or "Extremely Important"), and mean scores were calculated for each question across the set of respondents. Assigning a score of 0 to neutral perceptions was felt to be helpful in the analysis. Although there is ongoing debate on how best to analyse Likert data, there seems to be a general consensus in the research community that basic descriptive statistics (e.g., measures of mean and basic charts/graphs) are helpful, albeit analysing single items in isolation is not advised [47].

#### 4.1.1. General Information

Figure 1 shows the respondents' experience in BIM and GIS in rail projects. The data is further subdivided into BIM and GIS experience specifically in rail projects. The data is generally skewed to the left, with more respondents being relatively inexperienced. However, in moving from left to right in Figure 1, the balance shifts slightly from experience in GIS (dashed) to experience in BIM (solid). This may be due to the fact that BIM is arguably the more established technology, which is more applicable to the more frequent building projects (in contrast to the rarer rail projects).

Figure 2 indicates the years of experience using BIM and GIS together in an integrated way and shows that the greatest percentage had been implementing BIM and GIS in an integrated way for less than two years, but a sizeable minority (36.9%) had been implementing BIM and GIS in an integrated way for two years or more. This skew to the left may be because a rail project is a relatively infrequent megaproject and adopting new technologies takes time.

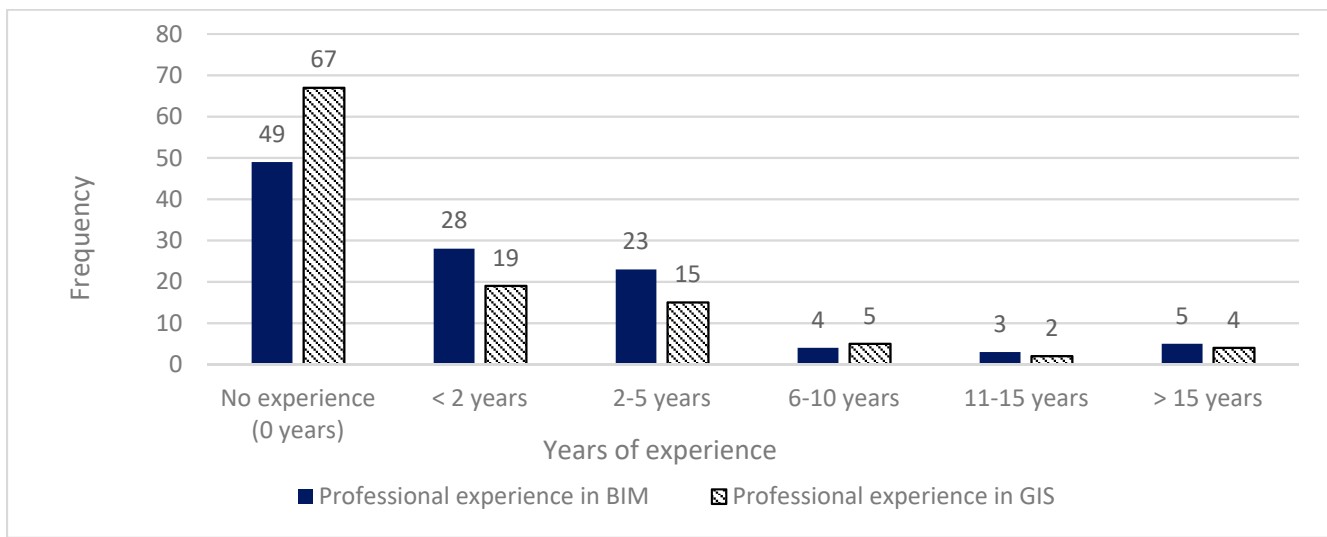

**Figure 1.** Years of experience of BIM and GIS.

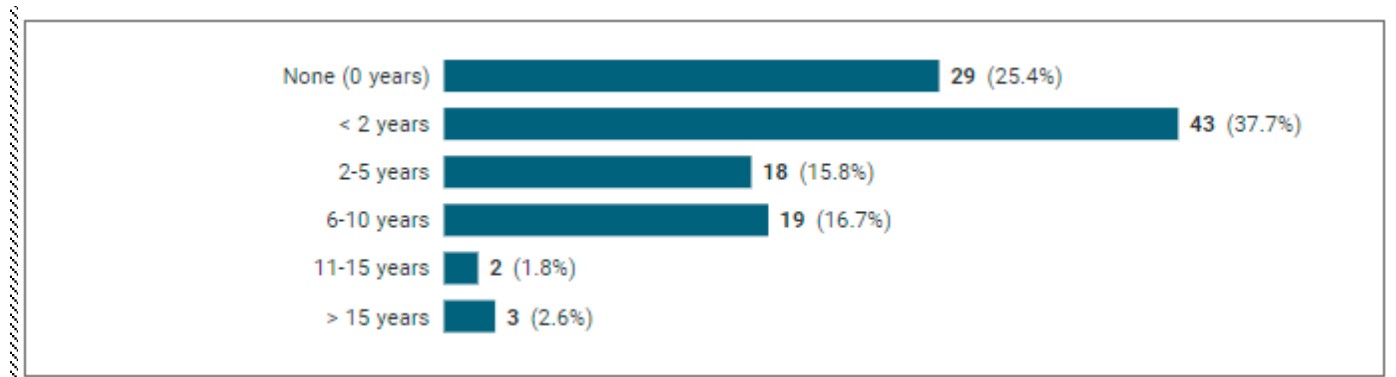

**Figure 2.** Years of implementing BIM and GIS in an integrated way.

Respondents were asked to report the size of their employer organisation and the procurement methods used. Figure 3 shows that the most common procurement methods were the traditional method and Design-Build, the former arguably less consistent with a BIM-based collaborative way of working. This finding could be because implementing new procurement mechanisms aligned with BIM and GIS is challenging due to lack of appropriate training, lack of awareness of BIM and GIS and the difficulty of encouraging stakeholders to change their existing working ways. A simple guide might encourage stakeholders to adopt these new technologies.

The questionnaire data indicated that respondents had more experience in BIM than in GIS. This may be because BIM is already or is becoming mandatory in several countries and BIM usage is wider than GIS. It may also be due to the wider applicability of BIM in a wide range of construction project types. It is also interesting that most respondents were self-taught in both BIM and GIS, as shown in Figure 4. This arguably demonstrates that there is a lack of suitable and affordable training, which may lead to the inappropriate implementation of BIM and GIS and poor realisation of the benefits from these tools.

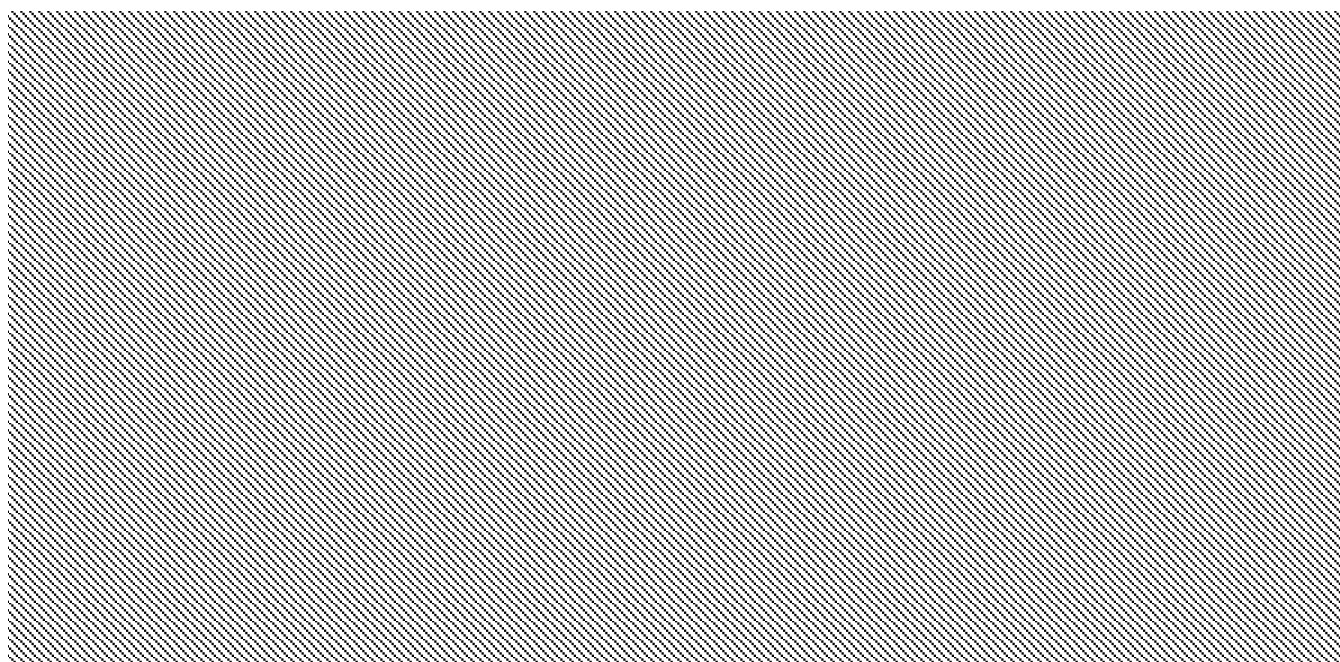

**Figure 3.** Procurement methods used.

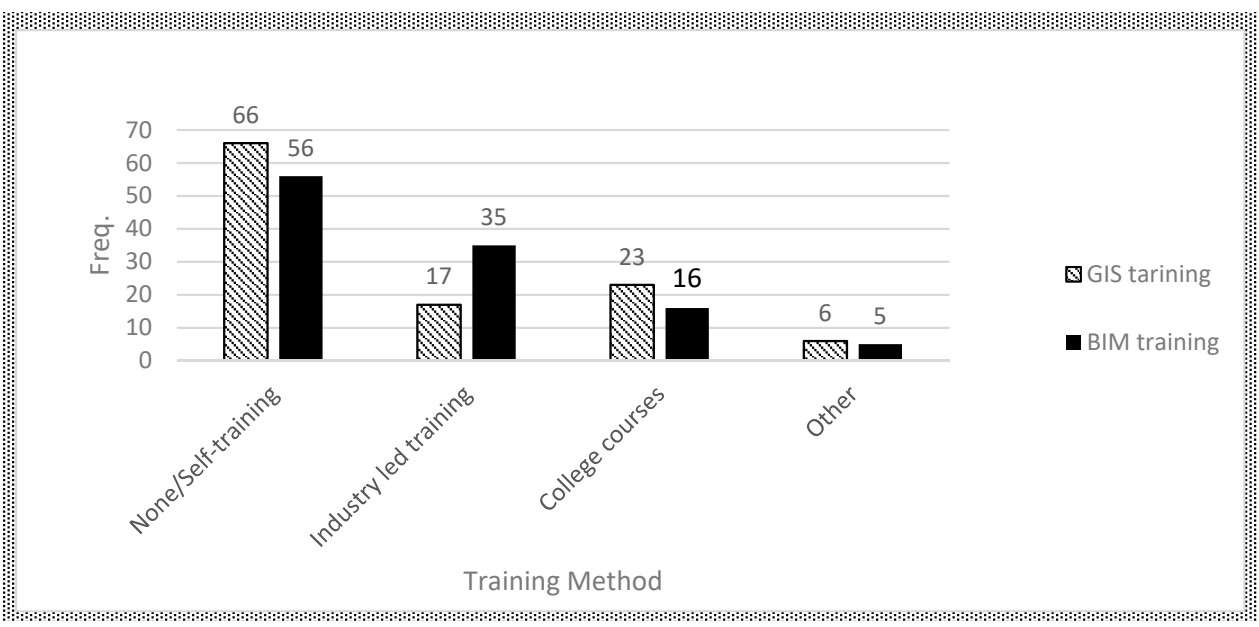

**Figure 4.** BIM and GIS Training Method.

A key issue when ascertaining how BIM and GIS can be integrated to support collaboration is to gauge the applicability of each technology across the project lifecycle. Figures 5 and 6 show the respondents' average assessment of the benefit of BIM and GIS at various project stages. Respondents were asked to rate the benefit of each technology at the stage, from -2 for "Not beneficial" to +2 for "Extremely beneficial". The scores were averaged across all respondents. Figure 5 shows that respondents felt that BIM is more beneficial for design and construction than for other stages. In contrast, GIS is beneficial for planning and pre-planning, while less beneficial for design, as shown in Figure 6. The complementary functionalities of the two technologies and the potential synergy from their integration have both emerged as themes from the literature review presented above. Figures 5 and 6 add an extra dimension to this, demonstrating that BIM and GIS between them appear to benefit whole project lifecycle.

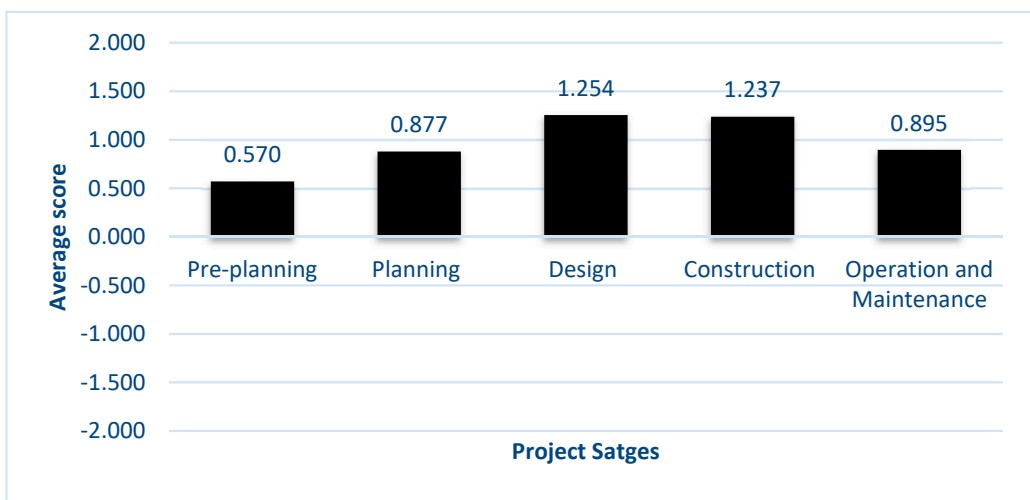

**Figure 5.** Benefit of BIM in various rail project stages.

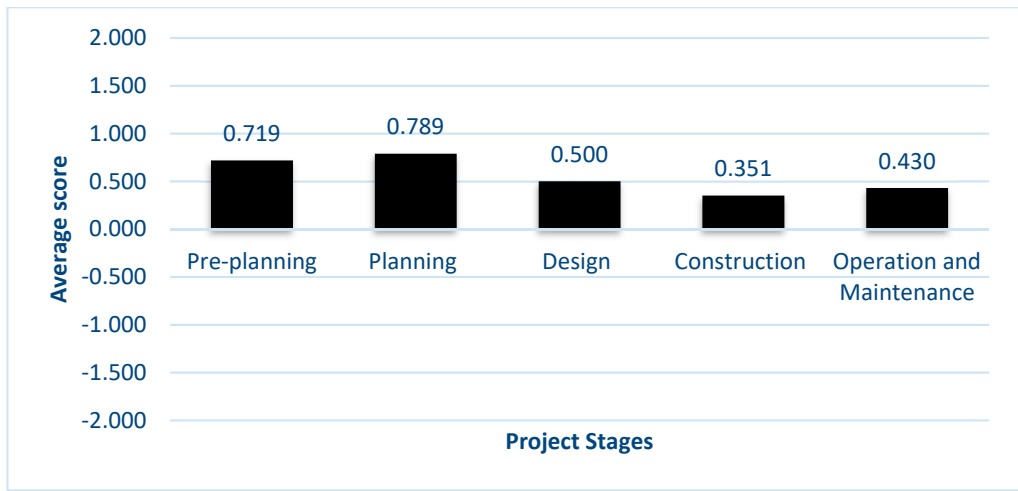

**Figure 6.** Benefit of GIS in various rail project stages.

4.1.2. Software Use

The questionnaire asked respondents to indicate the BIM and GIS software platforms they used in their professional practice. The data show that AutoCAD and Revit are about of equal usage, with a small difference unlikely to be statistically significant. ArcGIS was decisively the most common GIS platform.

4.1.3. Benefits of BIM and GIS in Isolation and Integrated Together

Respondents were asked to rate various benefits of BIM and GIS which were reported in the literature, as shown in Tables 1 and 2 respectively, from -2 for "Not beneficial" to +2 for "Extremely beneficial". From Table 1, the statement "BIM helps to detect clashes" was ranked top as the most important benefit realised by BIM, followed by "BIM supports better decision making".

For GIS, on the other hand, "GIS supports better decision making" was the highest-ranking benefit, followed by "GIS improves data availability", while "Improves the design quality" came third for both BIM and GIS, as shown in Table 2.

**Table 1.** Benefits of BIM.

| Rank | Statement | Mean | Std. Deviation |
|------|-----------|------|----------------|
| 1 | BIM helps to detect clashes | 1.3 | 1.053 |
| 2 | BIM supports better decision making | 1.3 | 0.976 |
| 3 | BIM Improves the design quality | 1.3 | 1.037 |
| 4 | BIM helps to avoid redesign issues | 1.2 | 1.016 |
| 5 | BIM improves data availability | 1.2 | 1.024 |
| 6 | BIM supports collaboration | 1.2 | 1.105 |
| 7 | BIM Improves productivity of estimator in quantity take-off | 1.2 | 1.05 |
| 8 | BIM reduces overall duration | 1.0 | 1.07 |
| 9 | BIM helps to reduce risks | 1.0 | 1.04 |
| 10 | BIM reduces overall cost | 1.0 | 1.131 |
| 11 | BIM supports- project delivery | 1.0 | 1.11 |

**Table 2.** Benefits of GIS.

| Rank | Statement | Mean | Std. Deviation |
|------|-----------|------|----------------|
| 1 | GIS supports better decision making | 0.8 | 1.299 |
| 2 | GIS improves data availability | 0.7 | 1.28 |
| 3 | GIS Improves the design quality | 0.5 | 1.294 |
| 4 | GIS helps to reduce risks | 0.5 | 1.23 |
| 5 | GIS supports collaboration | 0.5 | 1.329 |
| 6 | GIS supports- project delivery | 0.5 | 1.237 |
| 7 | GIS helps to avoid redesign issues | 0.4 | 1.286 |
| 8 | GIS reduces overall cost | 0.4 | 1.213 |
| 9 | GIS helps to detect clashes | 0.4 | 1.303 |
| 10 | GIS Reduces overall duration | 0.4 | 1.237 |
| 11 | GIS Improves productivity of estimator in quantity take-off | 0.2 | 1.364 |

It can be said that "information" is a common theme among these factors. Delivering the right information at the right time to the right person will lead to earlier clash detection, effective decisions, and avoidance of rework. Thus, BIM and GIS are not just general repositories of information, but tools which facilitate the routing of relevant information to specific recipients for specific purposes.

Respondents were asked to rate the potential benefits of BIM-GIS integration with respect to various design-related issues emerging from the literature. Ratings were scored from −2 for "Strongly Disagree" to +2 for "Strongly Agree". Figure 7 shows the scores averaged across all respondents. At the design stage, integrating BIM and GIS benefitted "Collaboration" in the first place, followed by "Visual exploration of design", "Quality of design" and "Information exchange and knowledge sharing and awareness of project partners (stakeholder)", with the latter two achieving the same average score. It is noteworthy however that all scores were comparable.

Figure 8 shows the respondents' rating of the importance of various barriers to BIM-GIS integration, from −2 for "Not important" to +2 for "Extremely important", averaged over all respondents. Paradoxically, collaboration stands out as the most important challenge, implying that collaboration is both a benefit of and a barrier to BIM-GIS integration.

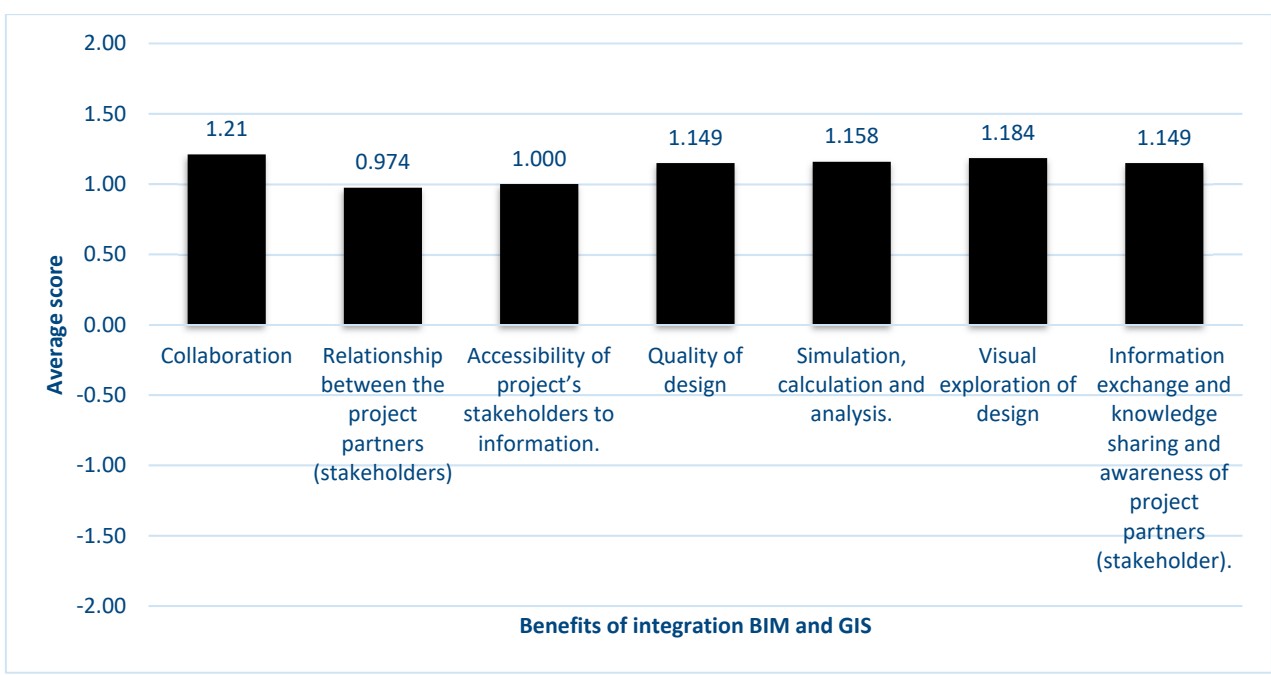

**Figure 7.** Benefits of Integration BIM and GIS in Design Stage.

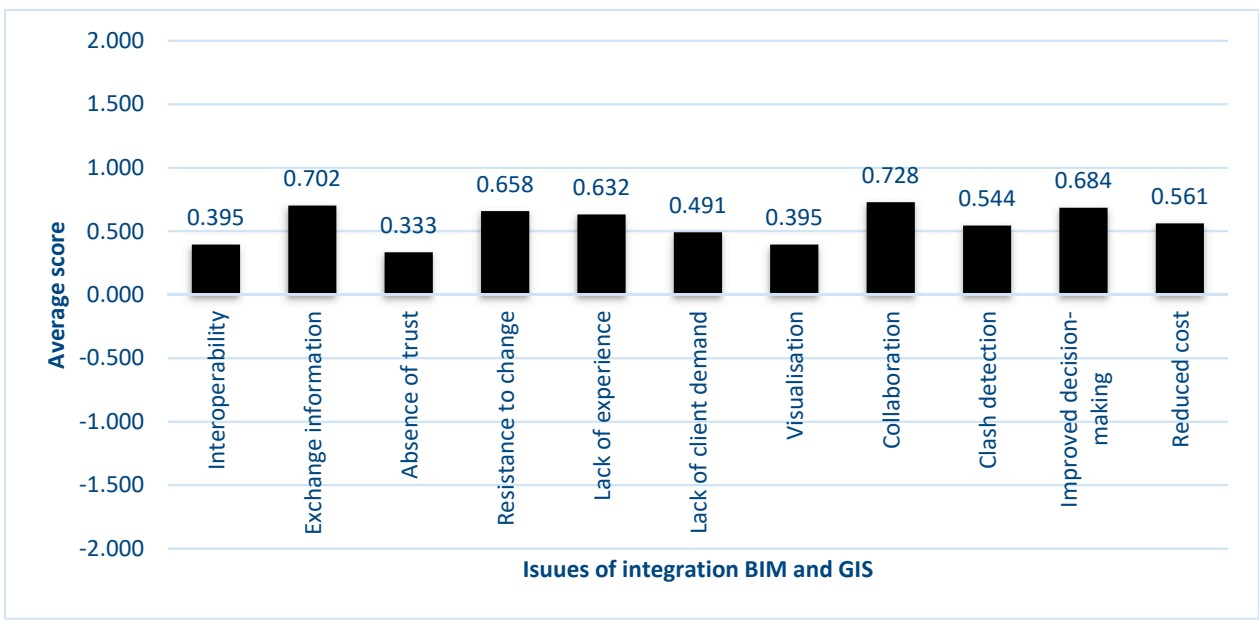

**Figure 8.** Barriers to Integrating BIM and GIS in the Design Stage in Rail Projects.

In summary, this survey yielded interesting results. Firstly, professionals are willing to learn new technologies (BIM and GIS), albeit through self-learning. Integrating BIM and GIS offers benefits and opportunities for projects. The stages in which BIM is most used are the design and construction stages while planning and pre-planning were the stages during which GIS is most used. This further highlights the complementarity of BIM and GIS. Furthermore, integration enhances coordination, collaboration, visualisation, clash detection, and decision making. The challenges facing this integration are the difficulty of collaboration, exchange of information and resistance to change. Effective collaboration would enable stakeholders to share, manage, and align decisions towards the shared goal. The subsequent interviews were used to explore issues of collaboration in more depth.

*4.2. Interview Analysis*

The purpose of the interviews was to explore in more depth the collaboration issues during the design stage of rail projects and the role of BIM and GIS in addressing these issues.

Twelve interviews were conducted via videoconferencing and three interviews were conducted in person (Table 3 lists the interviewees). The interviews were audio-recorded with each participant's permission and transcribed. Thematic analysis was used to draw out challenges to and suggestions for effective collaboration.

**Table 3.** Interviewee Codes.

| Interview Code | Years of Experience | Position |
| --- | --- | --- |
| I-1 | 11 | Head of BIM at a constructor |
| I-2 | 5 | Civil Engineer working for a small consultant |
| I-3 | +15 | Manager at a General contractor |
| I-4 | +5 | BIM Consultant at railway company |
| I-5 | +30 | Head of BIM at railway company |
| I-6 | 15 | BIM and GIS Manager at railway company |
| I-7 | 6 | BIM Director/ Head of GIS at railway company |
| I-8 | 20 | Engineering Information Manager at railway company |
| I-9 | 18 | Engineer at a general contractor |
| I-10 | 7 | BIM Engineer at railway company |
| I-11 | 4 | Architect at Architecture and Construction Management |
| I-12 | 8 | Senior Quality Control Engineer at a construction company |
| I-13 | 23 | Assistant Professor of Railway Engineering |
| I-14 | 12 | BIM specialist, senior civil /highway/infrastructure design engineer, Autodesk Certified Instructor |
| I-15 | +12 | Creative Director/Project Manager |

4.2.1. Collaboration Issues

The interviews highlighted many benefits of collaboration. For example, collaboration facilitates clash detection, enabling all the parties involved to work on the same piece of design at the same time (interviewees I-3 and I-9). Furthermore, through collaboration, the process of decision making will be effective and timely as all parties are engaged in the decision-making process (interviewee I-4). Moreover, collaboration unifies the "language" used, which means that all the participants should use the same file format and the same tools to exchange information, minimising information loss (interviewee I-5 and I-4). Ultimately, all interviewees agreed that collaboration leads to better outcomes in terms of time, reducing the project cost by avoiding rework, and enabling an effective decision-making process.

Despite its benefits, there are many significant challenges facing the process of effective collaboration. The most significant challenge is the lack of a clear process: how to collaborate, what information is needed, who is/are the right person(s) to whom to deliver this information, and when is the most appropriate time to exchange information (interviewee I-4). Interviewees I-12, I-4 emphasised that it is essential to identify the information needed, the latter stating:

"It is important to define the nature of the information needed. For example, what information needs to be imported from GIS to the BIM model. GIS can contain a lot of information which will never be needed over the project's life."

(Interviewee I-4).

Interviewee I-3 added that the aims of BIM and GIS need to be defined, and information transfer protocols need to be established for information exchange between BIM and GIS, in addition to setting out formally how participants in the design process should collaborate.

Similarly, I-5 highlighted that to collaborate effectively, several procedures need to be considered at the outset, such as establishing the information needed: "from the beginning of the project, you need to identify what information you need at any particular time, so you can make sure you get that information and use it to make decision at the end". This challenge can again be summarised as getting the right information at the right time for the right purposes and to the right person. Most standards currently in use do not provide this level of process transparency which is required (interviewee I-4). The latter identified Employers Information Requirements (EIR) and BIM Execution Plans (BEP) as mechanisms for addressing this but reiterated that further precision was necessary for achieving collaboration.

Interviewees I-5 and I-4 argued that collaboration requires a common language, but people tend to resist such standardisation of technology and language, stating that:

> "People are unfamiliar with new technologies and protocols. New technologies need to be explained to them in order to encourage them to adopt these new technologies. To collaborate they should use the same language. If people have a different language and refuse to use a common language it will be difficult to collaborate. So, it is about people accepting to collaborate".

In the same vein, I-8 (Engineering Information Manager at a railway company) argued that another significant challenge to collaboration is resistance to change. People tend to use their own software package. Raising the same issue, interviewee I-5 suggested that employees need to be encouraged to change their practices and to use new technologies by providing them with guidance on the collaboration process and freely providing the software required for collaboration.

The reasons for resistance to change in implementing new technologies may return to issues such as loss of data during information exchange or poor interoperability (interviewees: I-5, I-7, I-6, I-4). Interviewee I-11 also noted that "Collaboration between various stakeholders using different technologies is a challenge". This aligns with the views of Interviewees I-6 and I-4 when they emphasised that interoperability is the most common challenge to effective collaboration.

From the interview analysis, four main challenges can therefore be distilled as follows:

a. Collaboration is difficult to define, and people cannot agree a shared understanding of collaboration.
b. Information management: exchanging the right information, at the right time, with right people, for a particular purpose.
c. Resistance to adopting new technologies.
d. Interoperability between software tools.

### 4.2.2. Suggestions to Effective Collaboration

From analysing the interview data, several effective solutions can be suggested to collaborate effectively and to deliver the right information at the right time, which is the biggest challenge faced by project participants during the design stage. There was one overarching theme encompassing all suggestions for improved collaboration which emerged from the interviews: the need for a precise, collaboration-focused plan of work. For instance, interviewees I-1, I-5 and I-6 all indicated the need for a collaboration-based plan of work in order to produce a process model which precisely specifies collaboration activities in terms of the BIM and GIS information exchanges.

On the other hand, interviewees I-3 and I-4 emphasised the importance of having a clear BEP and the necessity of following an EIR. Interviewee I-2 suggested that early use of modelling methods can feed into system definition and effective use of GIS to aid in the integration of rail projects into the wider environment and the wider railway system.

Although there are several plans of work currently in circulation, it was thought worthwhile to develop a bespoke collaboration-based plan of work specifically for rail projects. All the participants were familiar with various plans of work and standards such as RIBA Plan of Work, BS 1192:2007, PAS 1192-2: 2013 and CIC Protocol. However, few of the participants mentioned following any of these. They all agreed that establishing a bespoke plan of work focusing on collaboration and specifically for rail projects would fill an important gap.

### 4.3. Summary of Findings and Development of "Collaborative Plan of Work"

The data highlights the potential of BIM and GIS. Professionals are willing to self-learn these technologies and adopt them (albeit "general resistance to new technologies" emerged as a challenge). The two tools are complementary and so their integration is worth pursuing. Collaboration is nebulous, but information exchange is a useful lens for its consideration. A dedicated workplan which sets out the required information exchanges through BIM-GIS integration would faceplate effective collaboration.

In order to develop a collaboration-based plan of work, the RIBA Plan of Work and GRIP Stages were chosen as the points of departure. Each of these plans of work has its strengths but neither of them focuses on collaboration. The RIBA Plan of Work addresses coordination but does not directly cover collaboration [8], while GRIP Stages is a scheme for managing investment to reduce and alleviate risks related to project delivery (NetwokRail, 2018). Furthermore, the overall approach of the GRIP Stages is driven by aspects of product rather than process [50]. Nevertheless, GRIP Stages has very specific features related to the railway to ensure the optimum design option is chosen and is feasible. The CPW was formulated by superimposing the RIBA Plan of Work and the GRIP Stages side-by-side, as shown in Table 4. The right-hand column of Table 4 sets out the tasks and outputs of the new, combined Collaborative Plan of Work, focusing on collaboration during the design phase of rail projects. The right-hand column of the table therefore summarises the original contribution of this work. This CPW is intended to fulfil the collaboration requirements left unmet by the two existing, distinct yet complementary plans, as noted by interviewees: I-4, I-5, and I-6. The CPW focuses on the collaborative process and information management among project participants to facilitate the design process and critical decision making.

**Table 4.** Developed Collaborative Plan of Work (CPW).

| RIBA work Plan | | | | GRIP Process | | | | | CPM, From This Research | | |
|---|---|---|---|---|---|---|---|---|---|---|---|
| Phase | Stage | Task | Output | No. | Stage | Task | Output | No. | Stage | Task | Output |
| Preparation | 0. Strategic Definition (Appraisal) | Identify the needs and objectives of the client, business case and potential constraints on development. prepare feasibility studies and options assessment to assist the client to decide to proceed or not | Clients requirements and preferable feasibility option | 1 | Output Definition | Define Project Output | Identify the definitions of the needs and requirements | 0 | Undertake Strategic Definition | Define Public Needs, Project objectives, business case, prepare feasibility study. (managing project need | Clients requirements, project objectives, feasibility study. (project need) |
| | 1. Preparation and Brief | Develop and confirm Initial Statement of requirements into the initial project brief | preferable feasibility option and project objectives | 2 | Pre-Feasibility | Define the investment scope, identify the constrains on the network, confirmation regarding that the output can be delivered economically and aligned with network strategy | Identify solutions for the requirements | 1 | Prepare Project Brief | Identify network constrains, Develop and confirm Initial Statement of requirements into the initial project brief. (managing information and project outline) | BIM execution plan, GIS execution plan, Designer responsibilities, specifications. (project outline) |
| | 2. Concept Design | Implement initial project brief and prepare concept design. The preparation of design concept includes proposals outline for structure and building services systems, specifications outline and plan of cost. procurements route review | Prepare Sustainability Strategy, Risk Assessments. Review and update the Project Execution Plan. | 3 | Option Selection | Address the constrains by developing options, assessing the options to select the optimum. Confirm that the output can be delivered economically | Determine single option, stakeholder approval. | 2 | Option selection development | Investigate to identify the options and develop it considering the economical delivered. Prepare concept design. (collaboration to make a decision) | Optimum layout of railway track, civil engineering structures, and systems. |
| | 3. Developed Design | Develop concept design and complete project brief | Concept Designs | 4 | Single Option Development | Developing the selected to single option Finalise business case and schedule implementing resources | Outline design | | Developed concept design | Preparing an outline of the concept design such as structures, civil, systems, and services plan of cost. (collaboration and using of technologies) | The final project brief, outline design of track, civil engineering structures, and systems |

**Table 4.** *Cont.*

| RIBA work Plan | | | | GRIP Process | | | | | CPM, From This Research | | |
|---|---|---|---|---|---|---|---|---|---|---|---|
| Phase | Stage | Task | Output | No. | Stage | Task | Output | No. | Stage | Task | Output |
| | 4. Technical Design | Prepare technical design, cost information, project strategy and specifications | technical designs cost information, project strategy and specifications | 5 | Detailed Design | Produces a complete robust engineering design to provide final estimation of cost, time, resources and risks. | Final design | | Developed detailed design | Prepare an outline of the technical design of the track, civil, systems in detail. (collaboration and using of technologies) | Detailed design of track, civil engineering structures, and systems. Construction strategy. Sustainably strategy |
| | 5. Construction | Manufacturing and constructing accordance with the construction programme and design queries | Project ready for operation. | 6 | Construction, Test and commission | Deliver to the specification and testing to confirm the workability of the asset and system in accordance with their design. | Project built, tested and authorised into use. | 3 | Construction | Manufacture and construct taking in consideration the construction programme and design queries | The project built and ready for operation. |
| Handover | 6. Handover and close out | Handover activities carried out | Conclude the The building contact | 7 | Project closeout | Settle the contractual accounts and put the warranties into their place also, carry out the benefits assessments | Project and project support system formally closed | 4 | Handover and project close-out | Settle the contractual accounts | The project formally closed, conclude the contracts |

## 5. Discussion

The aim of this paper is to identify and articulate collaboration requirements during rail design, recognising the role that an integrated application of BIM and GIS can play in meeting those requirements.

The questionnaire results represent the status of BIM and GIS and their use in rail projects. The results indicate, for example, that there is a lack of experience of BIM and GIS in rail projects. The application of both BIM and GIS has so far focused on buildings (73.1%) rather than infrastructure (12.2%) according to the research conducted by Ma and Ren [37]. Additionally, the questionnaire findings reveal a lack of training in BIM and GIS which leads to resistance to change due to lack of awareness as reported by others [21]. This is compatible with recent research [26] where it was concluded that lack of training is considered to be one of the challenges to achieving digital collaboration.

Furthermore, the questionnaire data revealed the most popular software packages used as AutoCAD and Revit for BIM and ArcGIS for GIS. This is consistent with the findings from Ma and Ren [37], who reported that similar platforms were used for BIM and GIS. Nevertheless, there appears to be limited use of packages more related to infrastructures (notably rail as a linear asset) such as Infraworks and QGIS [37].

The questionnaire also identified the project stages where BIM and GIS are the most useful. The findings align with previous studies [37] that BIM is used at the design stage while GIS is used mostly during the planning stage. The complementary functionality of the two sets of tools emerged repeatedly in this research, which aligns with previous studies. Zhu et al. [34] and Ma and Ren [37] observe that BIM provides a 3D model which can be used throughout the lifecycle of construction projects, while GIS is used to analyse and visualize problems related to location in geospatial science, environmental science, and natural resource management. Their complementary functionalities are a clear indication of the potential of their integrated use to support collaboration.

Of the questionnaire respondents in this study, a majority of 37.7% had integrated BIM and GIS for less than two years. This coincides with results from recent studies such as [34] which indicated that research on integrating BIM with GIS grew from only 3 studies in 2009 to 313 studies in 2017. This reflects the significance of this area and the growing interest amongst researchers. Realising the collaboration potential which arises from the complementary functionality of BIM and GIS and their integrated application requires a more detailed study of the requirements of the collaboration process during design. This was the focus of the interviews reported here.

The interviews revealed the major issue of information management during collaboration. Guidance is needed to support professionals in delivering the right information to the right collaborator at the right time. This aligns with the questionnaire results in this study, as well as previous studies which identified issues in collaboration and digital collaboration [17,26,51]. In particular, Anumba et al. [51] focus on collaborative information management and propose a conceptual framework based on semantic web technology. Such related studies support the finding reported here that a prescriptive process model of collaboration would provide a safeguarding framework, ensuring that the right information is delivered to the right person at the right time. Finally, collaboration requires guidelines and awareness [52]. BIM and GIS are recent complementary technologies whose integration would enable collaboration.

The developed CPW is focused on collaboration, deliverables and the information needed. Existing plans (e.g., RIBA, CIC and BSRIA Design Framework for Building Services) focus on the deliverables and to some extent the activities needed to produce these deliverables [53]. The Government Soft Landings (GSL) scheme aligns with RIBA Plan of Work and is developed to champion better outcomes for the built assets in the UK during the design and construction phase to ensure achievement of value throughout the operational lifecycle of an asset [54]. The GSL scheme was notably set out to be powered by BIM. The PAS1192-3 process [55], similarly to the CPW, emphasises information management, but is focused on the operational phase of an asset.

## 6. Conclusions and Recommendations

Effective collaboration is essential to the success of any project, particularly megaprojects such as rail projects. Despite the importance of collaboration, there are many challenges to its fruition; the most important one is providing the right information to the right person at the right time for the right purposes. To address this, this paper identified the requirements necessary for effective collaboration. The questionnaire findings highlighted the complementary roles of BIM and GIS in the project lifecycle and the potential of their integration for supporting collaboration. A follow-up round of interviews underlined information management as a crucial issue and highlighted the need for a precise collaboration-focused process to support the delivery of the right information to the right person at the right time. This paper presents a novel Collaboration Plan of Work based on the RIBA Plan of Work and GRIP Stages to define the process of effective collaboration for each stage of rail projects in terms of inputs, outputs and information needed. Future research will further develop and validate the CPW into a detailed and precise process model for collaboration based on an integrated application of BIM and GIS to rail projects.

The CPW is significant because it can potentially have a huge impact on the effective delivery of rail projects. This comes at a time when there is a flurry of high-profile rail projects around the world [56]. Once developed into a precise process model, the CPW can be used to configure the Common Data Environments used in the management of rail projects. The workflows and information flows facilitated by such configured platforms would foster collaboration for more effective delivery of these rail projects. The configuration of Common Data Environments is the basis for the operationalisation, validation and exploitation of the CPW, to be reported as part of future research.

A noteworthy limitation of this work is its geographical bias. The questionnaire respondents and interviewees were predominantly from the UK and the Middle East. The CPW is based on the RIBA and the GRIP Stages, which are both directed at the UK. This might limit the applicability of the CPW outside the UK.

**Author Contributions:** Conceptualization, S.K.; methodology, S.K.; formal analysis, S.K., P.D., K.B.B. and T.M.H.; writing—original draft preparation, S.K. and P.D.; writing—review and editing, S.K., P.D., K.B.B. and T.M.H. All authors have read and agreed to the published version of the manuscript.

**Funding:** This research received no external funding.

**Institutional Review Board Statement:** This research was conducted in compliance with the procedure set out by the Ethics Approvals (Human Participants) Sub-Committee at Loughborough University.

**Informed Consent Statement:** Informed consent was obtained from all subjects involved in the study.

**Conflicts of Interest:** The authors declare no conflict of interest.

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
