# Peer review of "Collaboration through Integrated BIM and GIS for the Design Process in Rail Projects: Formalising the Requirements"

_infrastructures, doi:10.3390/infrastructures6040052_

Round 1

Reviewer 1 Report

While this paper addresses an important topic, it is the opinion of this reviewer that the manuscript needs substantial improvement in order to be accepted for publication. 

While results of survey and interviews do provide insights as to the practical need for better collaboration and how this may be achieved, the research effort needs to demonstrate how these findings positively enable collaboration - either through real-world trials, simulations, etc. The authors speculate, probably somewhat correctly, that their proposed improvements to existing collaboration processes will be useful; however, there is no proof of these benefits. Thus, in its current form the paper does not make a measurable, substantial case for how collaboration can be improved in rail projects.

Other comments:

  • On page 1, what particularly about rail projects make them more complex than other projects? It will be helpful to elaborate on some of these issues of complexity and scale in rail efforts that can be addressed either partially or wholly by improved collaboration techniques/approaches.
  • On page 11, the authors indicate that based on the results of the survey, integrating BIM and GIS offer “huge benefits”. Without actual proof, this qualifier might be misplaced.
  • It will be helpful to include a better synthesis of the results of the SME interview. Perhaps a table of diagram illustrating the key insights from the survey/interview process. The textual discussion is rather hard to follow.
  • Suggest annotating/highlighting the "Developed CPW table" to point out where exactly findings of this research augment/improve upon the existing RIBA and GRIP processes.

Author Response

REVIEWER1

Thank you for your detailed and insightful reviews.  We have attempted to address them as shown below.

While results of survey and interviews do provide insights as to the practical need for better collaboration and how this may be achieved, the research effort needs to demonstrate how these findings positively enable collaboration - either through real-world trials, simulations, etc. The authors speculate, probably somewhat correctly, that their proposed improvements to existing collaboration processes will be useful; however, there is no proof of these benefits. Thus, in its current form the paper does not make a measurable, substantial case for how collaboration can be improved in rail projects.

Thank you for this insightful feedback.  Indeed, the research team is currently “operationalising” the CPW by developing it into a precise collaboration process and hardcoding this process in a commercial Common Data Environment, by configuring the project workflow.  This configured CDE will be separately evaluated and published as part of future research.  For the purposes of this paper, this plan has been be signposted in the Abstract and the Conclusions in order to highlight the significance of the work.

On page 1, what particularly about rail projects make them more complex than other projects? It will be helpful to elaborate on some of these issues of complexity and scale in rail efforts that can be addressed either partially or wholly by improved collaboration techniques/approaches.

The first paragraph in the introduction first cites the complexity of rail projects.  A short paragraph has been added (currently the second paragraph) which elaborates on this complexity and cites an additional publication on the complexity of information management.  This leads to the existing next paragraph on information management.  An attempt was made to keep this new paragraph brief so as not to lengthen the paper unnecessarily. 

On page 11, the authors indicate that based on the results of the survey, integrating BIM and GIS offer “huge benefits”. Without actual proof, this qualifier might be misplaced.

Well noted.  The word “huge” has been deleted.

It will be helpful to include a better synthesis of the results of the SME(?) interview. Perhaps a table or diagram illustrating the key insights from the survey/interview process. The textual discussion is rather hard to follow.

Subheading 4.3 has been added before the CPW is presented.  “4.3 Summary of Findings and Development of CPW””.  A summary of the findings has been added in this new subsection.

Suggest annotating/highlighting the "Developed CPW table" to point out where exactly findings of this research augment/improve upon the existing RIBA and GRIP processes.

The discussing Table IV in the body text, the following sentence has been added: The right-hand column of the table therefore summarises the original contribution of this work. 

Reviewer 2 Report

A very interesting article. The relationship between GIS and BIM is an ongoing, developing topic of interest, so an article that measures this relationship in a specific area is of special interest to the scientific community, both from a quantitative and qualitative point of view. On the other hand, collaborative work on these topics is a good starting point and underpins the interest in the relationship between these fields.

In point 4.1.3. Benefits of BIM and GIS in isolation and integrated together, there is a reference error of a table not found.

Author Response

REVIEWER2

Thank you for this constructive and through review.  We have attempted to address your points as below.

A very interesting article. The relationship between GIS and BIM is an ongoing, developing topic of interest, so an article that measures this relationship in a specific area is of special interest to the scientific community, both from a quantitative and qualitative point of view. On the other hand, collaborative work on these topics is a good starting point and underpins the interest in the relationship between these fields.

Thank you for this positive feedback.

In point 4.1.3. Benefits of BIM and GIS in isolation and integrated together, there is a reference error of a table not found.

We apologise for this.  This should refer to Table I.  This broken reference is only visible following the automatic conversion to PDF and we have checked that it does not occur.

Reviewer 3 Report

Thank you for submitting your paper “Collaboration through Integrated BIM and GIS for the Design Process in Rail Projects: Formalising the Requirements” to the Journal of Infrastructures.

In my opinion, the topic is of great interest but needs some improvements before publication.

Most importantly, the authors should emphasize the real contribution of this work compared to recently published work. The novelty of this paper should also be well described and emphasized in the title, abstract, and conclusion. Please work on this and show us why this work is valuable.

I suggest reorganizing the abstract, highlighting the new features introduced, it should contain answers to the following questions:

- What problem was studied and why is it important?

- What methods were used?

- What results are important?

- What conclusions can be drawn from the results?

- What is the novelty of the work and where does it go beyond previous efforts in the literature?

Author Response

REVIEWER3

Thank you for the detailed and insightful review.  We have thoroughly revised the manuscript to address your feedback.

Thank you for submitting your paper “Collaboration through Integrated BIM and GIS for the Design Process in Rail Projects: Formalising the Requirements” to the Journal of Infrastructures. In my opinion, the topic is of great interest but needs some improvements before publication.

Thank you.  The paper has been revised throughout to emphasise the originality, significance and rigour of the work.

Most importantly, the authors should emphasize the real contribution of this work compared to recently published work. The novelty of this paper should also be well described and emphasized in the title, abstract, and conclusion. Please work on this and show us why this work is valuable.

Abstract: the word uniquely is now italicised to emphasize the novelty of the work.  The end of the abstract has been revised to emphasise the significance of the findings. 

Conclusions: the penultimate paragraph (before closing limitations paragraph) already attempts to make a bold statement of the significance of this research.  The beginning of that paragraph has been revised slightly, and the term significance used for emphasis.

I suggest reorganizing the abstract, highlighting the new features introduced, it should contain answers to the following questions:

- What problem was studied and why is it important?

- What methods were used?

- What results are important?

- What conclusions can be drawn from the results?

- What is the novelty of the work and where does it go beyond previous efforts in the literature?

The abstract has been advised. The beginning now clearly articulates the problem addressed: problems with delivery of rail projects, traced back to issues of collaboration.

The methods were already clearly articulated.

The end of the abstract is revised to highlight the novelty and significance of the results/conclusions.

Round 2

Reviewer 1 Report

Thank you for the detailed responses as to how each comment was addressed. This reviewer has no further comments.